# Demographic differentials of lung cancer survival in Bangladeshi patients

**Muhammad Rafiqul Islam**[1]*, **A. T. M. Kamrul Hasan**[1], **Nazrina Khatun**[1], **Ishrat Nur Ridi**[2], **Md. Mamun Or Rasheed**[3], **Syed Mohammad Ariful Islam**[4], **Md Nazmul Karim**[5]

1 Department of Medical Oncology, National Institute of Cancer Research and Hospital, Dhaka, Bangladesh, 2 Department of Medicine, Dhaka Medical College and Hospital, Dhaka, Bangladesh, 3 Department of Oncology, Bangabandhu Sheikh Mujib Medical University, Dhaka, Bangladesh, 4 Department of Medicine, Kurmitola General Hospital, Dhaka, Bangladesh, 5 School of Public Health and Preventive Medicine, Monash University, Melbourne, Australia

* alongsinger@gmail.com

## Abstract

### Background

Lung cancer is the leading cause of cancer-related mortality worldwide. Demographic differential has been linked with the treatment outcome and survival in recent literature, mostly from the developed world. Considering diversity in population characteristics across income strata, it's worth assessing the link in low- and middle-income population as well. Current study aimed to assess the association of demographic characteristics with lung cancer survival in Bangladeshi lung cancer patients.

### Methods & results

All newly diagnosed primary lung cancer cases attending the national institute of cancer research & Hospital (NICRH), a tertiary cancer care center in Dhaka, Bangladesh between 2018 and 2019 were considered for the study. Demographic information and clinical data were obtained from the patients' medical records by a trained physician. Survival estimate was generated using the Kaplan-Meier method and compared across demographic and clinicopathological categories using the log-rank test. Hazard ratio and 95% CI for treatment options are generated fitting multivariable Cox proportional hazard regression.

Among 1868 patients, 84.6% were males and 15.4% were females, average (± standard deviation) age at diagnosis was 59.6±10.9 years, only 10.8% had not consumed tobacco of any form. Around two-thirds of the patient had Eastern Cooperative Oncology Group (ECOG) performance score ≥2, 29.5% had at least one comorbidity and 19.4% had metastasis at the time of presentation. Higher survival was associated with institutional education (HR 0.9; 95% CI 0.77, 0.99), and receipt of combined radiotherapy and chemotherapy (HR 0.56; 95% CI 0.46, 0.65; p <0.001). In contrast, lower survival was associated with older age between 60–69 years (HR 1.3; 95% CI 1.3, 1.5;), age ≥ 70 years (HR 1.4; 95% CI 1.1, 1.7), having any comorbidity (HR 1.1; 95% CI 1.0, 1.3), with ECOG score ≥ 3 (HR 1.41; 95% CI 1.01, 1.96) and receipt of radiotherapy treatments only (HR 1.6; 95% CI 1.3, 1.9).

**Data Availability Statement:** Data used in this study are owned by a Medical oncology department of NICRH and contains potentially identifying patient information. The data can be

requested from Assistant Professor of the Department Medical oncology, Dr. Ferdous Ara Begum, email: dr.ferdousara1963@gmail.com (office address: Room no. 421, 3rd floor, Academic Building, National Institute of Cancer Research and Hospital, TB gate Mohakhali, Dhaka, Bangladesh).

**Funding:** The authors received no specific funding for this work.

**Competing interests:** The authors have declared that no competing interests exist.

## Conclusion

Older age, presence of one or more comorbidity, poorer performance status, and treatment with only RT appeared as a significant predictor of poorer prognosis of lung cancer in Bangladeshi patients. In contrast, having institutional education and treatment with combined Radiotherapy and Chemotherapy appeared as a predictor of a better prognosis. The finding of this study could serve as a basis for future studies inquiring into novel approaches for certain subgroups of patients believed to be challenged in limited resources.

## Introduction

Lung cancer is the leading cause of cancer-related mortality worldwide with more than one million deaths annually [1]. According to the GLOBOCAN 2018, lung cancer is the most often diagnosed malignancy (11.6% of the total incident cancer cases in 2018) with an age-standardized incidence rate of 22.5 (31.5 in male, 14.6 in female) per 100,000 person-years worldwide in 2018 [2]. Despite the improvement in diagnostic facilities and treatment modalities, the prognosis of the disease remains poor. Several factors, including, histopathological variety [3–5], stage at diagnosis [6], and treatment modality are linked with the prognosis as well as with the survival [6]. Among patient characteristics, performance status, comorbidity, and response to the treatment were also reported to influence patient survival [7,8]. Patient's demographic characteristics, such as gender [4,9,10] and race [11] have been linked with the treatment outcome and the survival in the recent literature, mostly from the western world. Despite, considerable evidence from the developed countries, the data is scarce in resource-limited countries like, Bangladesh. Few studies in Asian populations showed better responses to treatment and better survival outcomes among the non-smoker patients [12,13]. Considering diversity in population characteristics across income strata, it is worth assessing the link in low- and middle-income population as well. A small study in Bangladesh reported that, only 27% of the patients survive up to one year following diagnosis. The average survival is slightly longer in younger (<40 years) and female patients [14]. Further, research on the link of demographic characteristics of lung cancer patients in Bangladesh has the potential to facilitate better cancer management in the resource-limited setting. The aim of the study is to study the influence of demographic characteristics on treatment outcome and survival of lung cancer patients.

## Material & methods

Current study included all the newly diagnosed and histologically confirmed primary lung cancer (ICD-10-CM C34) patients, attended the medical oncology department of National Institute of Cancer Research & Hospital (NICRH), a tertiary care center at Dhaka, Bangladesh, during the year 2018 and 2019. NICRH is a public funded hospital, where treatments are provided free of cost, as a result this is generally the destination of the patient across the spectrum of the disease stages. However, despite being the apex referral cancer center in the country, NICRH is not yet equipped with the facility and capacity of targeted therapy, immunotherapy in particular. As a result, our patient population did not include patients with targeted therapy (i.e. immunotherapy).

Demographic information (age, gender, body mass index, education level, socioeconomic status (SES), smoking, and smokeless tobacco consumption status) were extracted from

patient records. Body mass index (BMI) was categorized into <18.5, (underweight), ≥18.5 and 25 (Normal weight) and >25 (overweight) [15]. Level of education was grouped into 'no formal schooling (illiterate)', '1 to 5 years of schooling (primary)', and '> 5 years of schooling (Secondary and above)'. Monthly family income was categorized into < \$115 (low income), \$115–\$235 (lower middle) and > \$235 (middle to upper). never or less than one year of smoking history defined as no smoker or no user of smokeless tobacco. The clinical data including date of diagnosis, anatomical site, histological types, comorbidity, performance status, and treatment modalities were obtained from the patients' medical records by a trained physician. A patient was considered to have co-morbidity if the patient has been suffering or receiving treatment for a major existing condition such as, diabetes, hypertension, heart disease, or chronic obstructive pulmonary disease, etc. Comorbidity data of the patient were collected by trained data collector (physician) from the, chart review, current medication record, and cancer treatment eligibility check-up record conducted at the center.

Performance status was assessed based on the Eastern Cooperative Oncology Group (ECOG) performance score 0 to 4 (0 = Fully active, 1 = Restricted in physically strenuous activity, 2 = Ambulatory and capable of all self-care but unable to carry out any work activities, 3 = Capable of only limited self-care and 4 = Completely disabled) which represent the patient's level of function and capability of self-care [16]. The patients were allocated into chemotherapy (CT) or radiotherapy (RT) as indicated following standard guidelines. After complete assessment of the patient, Histology based standard chemotherapy and radiotherapy protocols guided by the National Comprehensive Cancer Network (NCCN) guidelines were followed by an institutional multidisciplinary tumor board (comprise medical Oncologist, Radiation oncologist, surgical oncologist, radiologist and pathologist) for both curative and palliative setting. The patients were followed up (over the telephone) until the event of death or June 30, 2020, whichever came first and where the patient could not be contacted, telephonic contact of the patient's relatives (assigned at the recruitment) was conducted to ascertain the present status of the subject. Death data was further confirmed via the death registration department. The Ethical Review Board of the National Institute of cancer research and hospital approved the study protocol (ref no NICRH/Ethics/2020/124). All subjects gave written informed consent following the Declaration of Helsinki.

## Statistical analysis

Descriptive statistics were generated regarding patient demographics, clinical characteristics, and treatment parameters. Duration of survival was calculated from the date of confirmation of the diagnosis to date of death or last date of follow-up. When contact could not be established despite three attempts, patients were classified as lost to follow-up and were considered censored since the last follow-up contact.

Survival estimates were generated using the Kaplan-Meier method and compared across demographic and clinico-pathological (age, gender, body mass index, education, SES, tobacco use, site, histology, comorbidity, performance status, and treatment) categories using the log-rank test. Univariate and multivariable cox proportional hazards regression was fit to assess the association of demographic and clinico-pathological factors. Treatment outcome was assessed using multivariable cox proportional hazard regression across strata of predictors adjusting for all plausible confounders. Hazard ratio and 95% CI for the treatment options are generated fitting multivariable Cox proportional hazard regression adjusting for plausible confounders and considering "no treatment" as the reference category.

## Results

Among 1868 patients included in the study, 84.6% were males and 15.4% were females. The demographic and clinical characteristics of the patients are presented in Table 1. Average (± standard deviation) age at diagnosis was 59.6±10.9 years. Among them 13.9% were aged <50 years, 25.6% between 50–59 years, 38.2% between 60–69 years and 22.3% were aged >70 years. The majority (52%) of the patients were with normal weight, around 40% were underweight and 8.1% were overweight. Around two-third of the patients (66.2%) had no formal education. The majority (55.4%) of the patient had a family income of < $115 per month. Overall, more than eighty percent of patients were smokers and around half of the patients were smokeless tobacco users, "44.3% (n = 827) patients consumed both the form of tobacco" and only 10.8% had not consumed tobacco of any form. Gender specific tobacco exposure shows that tobacco consumption was higher among males (85.4%) in comparison to female (55.6%).

There was slight right-sided predominance (58.5%) anatomically. Overall squamous cell carcinoma (42.3%) was the predominant histological verity followed by Adenocarcinoma (38.3%), small cell carcinoma (11.8%), and undifferentiated or other (7.6%). Prevalence of Adenocarcinoma is higher in female (46.3%) than male (36.9%), and conversely the prevalence of Squamous cell carcinoma is higher in male (43.2%) than female (37.5). About 19.4% patients had metastatic disease during the presentation. A third of those patients had liver metastasis. Around 29.5% (n = 551) patients were reported to have at least one comorbidity, the comorbidities were Diabetes (9.0%), Hypertension (10.1%), COPD (11.9%), Asthma (1.0%), IHD (0.9%), Arthritis (1.7%). Out of the 551 patients with comorbidities, 461 had one comorbidity, 82 had two comorbidities and 7 patients had three or more comorbidities. Around 65.6% of the patient's ECOG performance scores were ≥2. Forty percent of patients did not receive any treatment, were unfit for further treatment, or dropped out from the treatment. Around one-third of the subjects were treated with only systemic chemotherapy (CT), 6.5% were treated with radiotherapy (RT) and 17.5% received both CT and RT.

Fig 1 illustrates the distribution of the survival estimates among lung cancer patients across demographic and clinical categories. The average survival was found significantly higher in younger age (<60 years), patients with any formal schooling, without comorbid illness, with lower ECOG (<2), and those received both CT and RT. The association was found to be consistent when assessed through univariate Cox proportional hazard regression (Table 2). Patients with institutional education (HR 0.88; 95% CI 0.77, 0.99; p 0.039), and those received both CT and RT (HR 0.56; 95% CI 0.46, 0.65; p <0.001) were found to have higher survival. Patients aged between 60–69 years (HR 1.26; 95% CI 1.06, 1.49; p 0.10) aged ≥70 years (HR 1.36; 95% CI 1.13, 1.65; p 0.001), having any comorbidity (HR 1.14; 95% CI 1.01, 1.29; p 0.023), with ECOG score 3 and 4 (HR 1.41; 95% CI 1.01, 1.96; p 0.042 and HR 1.82; 95% CI 1.26, 2.63; p 0.001 respectively), and those received RT only (HR 1.58; 95% CI 1.29, 1.93; p <0.001) were found to be have lower survival.

Treatment outcome of CT only, RT only and CT+RT combined assessed using multivariable cox proportional hazard regression across strata of predictors, considering "no treatment" as the reference category and adjusting for plausible confounders age, performance status, gender, BMI, SES, education, and histology (Table 3). Age was a significant predictor of mortality risk across all treatment categories. Among the patients received only CT and those aged <50 years showed survival advantage. But among the patients who received RT only, survival decreased with increasing age over 50, in contrast, among those who received both CT and RT, the survival advantage was seen across all age groups, particularly with increasing age. The RT and CT combined seem to provide similar survival advantages across strata of education,

**Table 1. Characteristics of demographic and clinic-pathologic variables.**

| Variables | Categories | Frequency | Percent (%) |
|---|---|---|---|
| **Age** | < 50 Years | 256 | 13.9 |
| | 50–59 years | 474 | 25.6 |
| | 60–69 years | 707 | 38.2 |
| | 70 Years and above | 412 | 22.3 |
| **Gender** | Male | 1,580 | 84.6 |
| | Female | 288 | 15.4 |
| **BMI**[*] | Normal | 876 | 51.9 |
| | Underweight | 674 | 40.0 |
| | Overweight | 137 | 8.1 |
| **Education** | Illiterate | 1,227 | 66.2 |
| | Primary | 224 | 12.1 |
| | Secondary & above | 403 | 21.7 |
| **SES**[*] | Low | 1,025 | 55.4 |
| | Lower middle | 519 | 28.0 |
| | Middle to Upper | 308 | 16.6 |
| **Smoking tobacco** | No | 358 | 19.1 |
| | Yes | 1,511 | 80.9 |
| **Smokeless tobacco** | No | 885 | 47.3 |
| | Yes | 984 | 52.7 |
| **Site** | Right | 1,078 | 58.5 |
| | Left | 766 | 41.5 |
| **Histology** | Adenocarcinoma | 704 | 38.3 |
| | Squamous cell carcinoma | 776 | 42.3 |
| | Small cell carcinoma | 217 | 11.8 |
| | Undifferentiated | 119 | 6.5 |
| | Others | 20 | 1.1 |
| **Comorbidity** | No | 1,318 | 70.5 |
| | Yes | 551 | 29.5 |
| **ECOG**[*] **performance score** | 0 | 60 | 3.2 |
| | 1 | 583 | 31.2 |
| | 2 | 773 | 41.4 |
| | 3 | 342 | 18.3 |
| | 4 | 110 | 5.9 |
| **Treatment** | None | 748 | 40.0 |
| | CT only | 673 | 36.0 |
| | RT only | 122 | 6.5 |
| | RT & CT | 326 | 17.5 |
| **Death** | No | 399 | 21.4 |
| | Yes | 1,468 | 78.6 |

[*]BMI = Body mass index, SES = Socioeconomic Status, ECOG = Eastern Cooperative Oncology Group.

socioeconomic status, BMI category, and Comorbidity. Small cell carcinoma showed to have the worst survival outcome with RT only but had better survival with CT and RT combined. Patients with adenocarcinoma had the best prognosis with combined treatment modality (CT and RT).

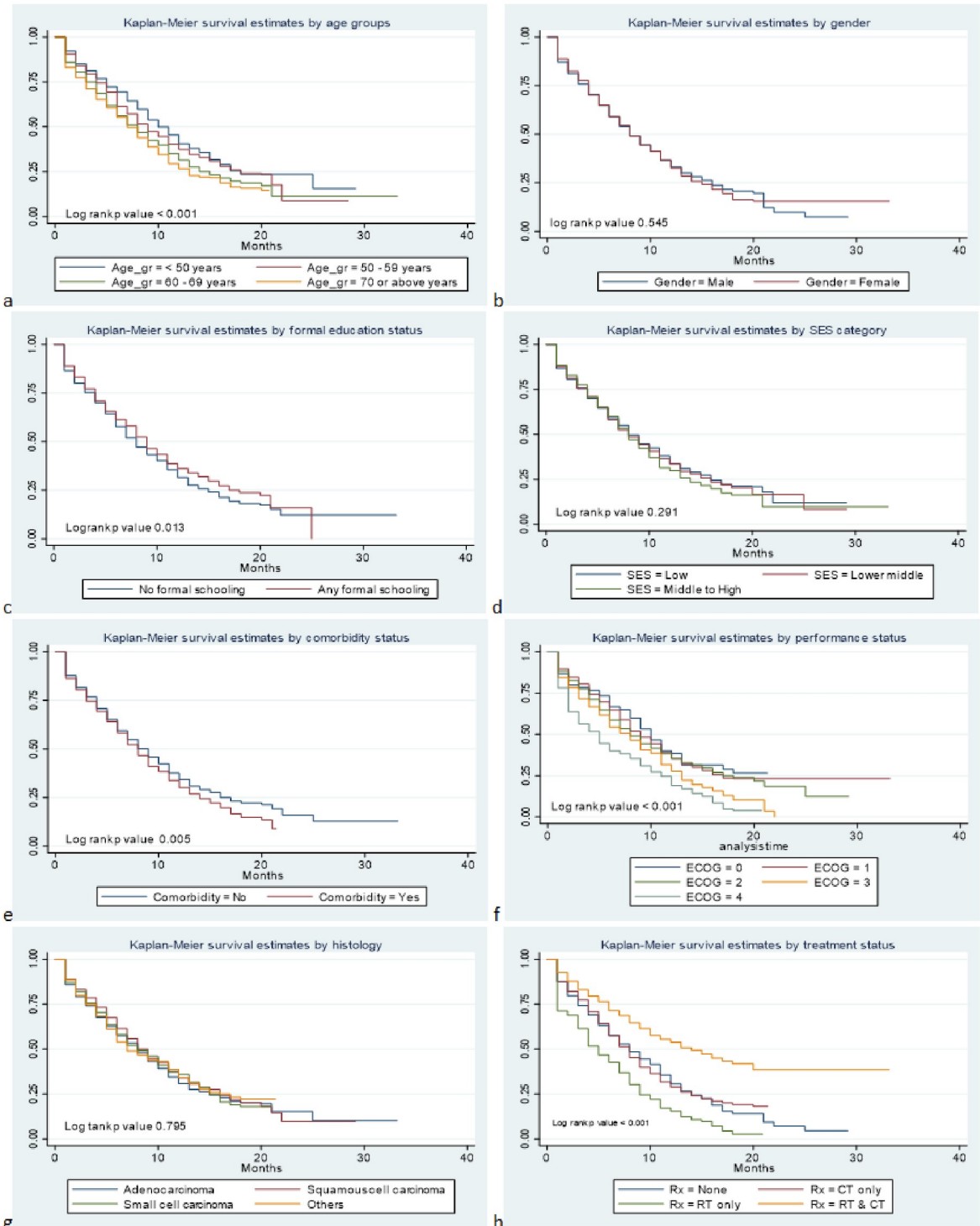

**Fig 1. Kaplan-Meier curves illustrating comparison of survival estimates across strata of age (a), gender (b), smoking status (c), SES (d), Comorbidity status (e), ECOG performance status (f), Histology type (g) and treatment type (h).** Statistical significance of difference was determined by log rank test p value<0.05.

**Table 2. Univariate and multivariable Cox-proportional hazards models for survival following among lung cancer patients attend at NICRH for treatment.**

| | Univariate cox regression | | Multivariate cox regression | |
|---|---|---|---|---|
| | HR (95% CI) | P value | HR (95% CI) | P value |
| Age | | | | |
| < 50 Years | Reference | | | |
| 50–59 years | 1.08 (0.90, 1.28) | 0.418 | 1.1. (0.92, 1.32) | 0.294 |
| 60–69 years | 1.28 (1.08, 1.50) | 0.004* | 1.26 (1.06, 1.50) | 0.009* |
| 70 Years and above | 1.40 (1.17, 1.68) | <0.001* | 1.37 (1.14, 1.65) | 0.001* |
| Gender | | | | |
| Male | Reference | | | |
| Female | 0.88 (0.78, 0.98) | 0.017 | 1.00 (0.88, 1.16) | 0.977 |
| Education | | | | |
| No formal schooling | Reference | | | |
| Any formal schooling | 1.12 (0.99, 1.26) | 0.078 | 0.87 (0.77, 0.98) | 0.032* |
| SES* | | | | |
| Low SES | Reference | | | |
| Lower middle SES | 1.04 (0.92, 1.17) | 0.570 | 1.05 (0.93, 1.20) | 0.418 |
| Middle to upper SES | 1.11 (0.97, 1.28) | 0.136 | 1.08 (0.92, 1.26) | 0.367 |
| Histology | | | | |
| Adenocarcinoma | Reference | | | |
| Squamous cell carcinoma | 0.95 (0.85, 1.06) | 0.373 | 0.92 (0.82 1.04) | 0.198 |
| Small cell carcinoma | 0.99 (0.83, 1.18) | 0.934 | 0.95 (0.79, 1.13) | 0.494 |
| Others | 0.95 (0.77, 1.17) | 0.621 | 0.94 (0.77, 1.17) | 0.590 |
| Comorbidity | | | | |
| No | Reference | | | |
| Yes | 1.16 (1.04, 1.30) | 0.007* | 1.14 (1.01, 1.28) | 0.034 |
| ECOG performance score* | | | | |
| 0 | Reference | | | |
| 1 | 1.08 (0.79, 1.48) | 0.633 | 1.09 (0.79, 1.51) | 0.587 |
| 2 | 1.12 (0.82, 1.52) | 0.488 | 1.12 (0.81, 1.53) | 0.506 |
| 3 | 1.46 (1.06, 2.01) | 0.020 | 1.41 (1.01, 1.96) | 0.042* |
| 4 | 1.95 (1.37, 2.79) | <0.001* | 1.82 (1.25, 2.63) | 0.001* |
| Treatment | | | | |
| None | Reference | | | |
| RT* only | 1.59 (1.31, 1.94) | <0.001* | 1.58 (1.29, 1.93) | <0.001* |
| CT* Only | 0.97 (0.86, 1.08) | 0.554 | 1.05 (0.92, 1.18) | 0.471 |
| RT and CT | 0.52 (0.45, 0.60) | <0.001* | 0.55 (0.46, 0.65) | <0.001* |

SES = Socioeconomic Status, ECOG = Eastern Cooperative Oncology Group, RT = Radiotherapy, CT = Chemotherapy.

## Discussion

Having institutional education and treatment with both CT and RT appeared as a predictor of a better prognosis. Treatment with CT and RT combined appear to have better survival for adenocarcinoma and squamous cell carcinoma. In contrast, older age, presence of comorbidity, poorer performance status, and treatment with only RT appeared as a significant predictor of poorer prognosis of lung cancer.

In our study, age appeared as a significant predictor of survival among lung cancer that coincides with contemporary literature [17]. With the increasing age, the incidence of chronic

**Table 3. Treatment outcome across socio-demographic and disease status strata.**

| | CT only# | | RT only# | | Both CT+RT# | |
|---|---|---|---|---|---|---|
| | HR (95% CI) | P value | HR (95% CI) | P value | HR (95% CI) | P value |
| Age | | | | | | |
| < 50 Years | 0.60 (0.43, 0.86) | 0.005* | 1.51 (0.73, 3.14) | 0.271 | 0.50 (0.31, 0.79) | 0.003* |
| 50–59 years | 1.02 (0.78, 1.32) | 0.881 | 1.88 (1.16, 3.03) | 0.010* | 0.66 (0.47, 0.91) | 0.012* |
| 60–69 years | 1.14 (0.93, 1.39) | 0.203 | 1.57 (1.10, 2.22) | 0.011* | 0.51 (0.38, 0.68) | <0.001* |
| ≥70 Years | 1.25 (0.97, 1.63) | 0.085 | 1.77 (1.20, 2.62) | 0.0004* | 0.46 (0.31, 0.68) | <0.001* |
| Gender | | | | | | |
| Male | 1.01 (0.96, 1.264) | 0.168 | 1.83 (1.44, 2.32) | <0.001* | 0.51 (0.42, 0.62) | <0.001* |
| Female | 0.74 (0.52, 1,03) | 0.077 | 1.27 (0.76, 2.12) | 0.357 | 0.74 (0.49, 1.18) | 0.153 |
| Education | | | | | | |
| Illiterate | 1.10 (0.95, 1.29) | 0.208 | 1.63 (1.27, 2.11) | <0.001* | 0.55 (0.44, 0.68) | <0.001* |
| literate | 0.90 (0.73, 1.13) | 0.367 | 1.54 (1.03, 2.29) | 0.035* | 0.52 (0.38, 0.70) | 0.001* |
| SES* | | | | | | |
| Low SES | 1.03 (0.88, 1.22) | 0.726 | 1.78 (1.34, 2.36) | <0.001* | 0.57 (0.45, 0.73) | <0.001* |
| Lower middle SES | 1.06 (0.83, 1.34) | 0.638 | 1.43 (0.92, 2.23) | 0.114 | 0.57 (0.42, 0.79) | 0.001* |
| Middle to upper SES | 0.89 (0.65, 1.22) | 0.470 | 1.35 (0.78, 2.33) | 0.290 | 0.37 (0.23, 0.59) | <0.001* |
| BMI* | | | | | | |
| BMI ≥18.5 | 1.00 (0.85, 1.18) | 0.983 | 1.65 (1.20, 2.24) | 0.002* | 0.59 (0.47, 0.73) | <0.001* |
| BMI < 18.5 | 1.09 (0.89, 1.33) | 0.391 | 1.59 (1.17, 2.14) | 0.003* | 0.48 (0.36, 0.63) | <0.001* |
| Histology | | | | | | |
| Adenocarcinoma | 1.06 (0.87, 1.30) | 0.548 | 1.30 (0.89, 1.89) | 0.168 | 0.48 (0.35, 0.65) | <0.001* |
| Small cell Ca | 0.91 (0.75, 1.11) | 0.384 | 1.92 (1.40, 2.62) | <0.001* | 0.50 (0.38, 0.65) | <0.001* |
| Squamous cell Ca | 1.42 (0.98, 2.06) | 0.067 | 1.89 (0.93, 3.74) | 0.073 | 0.75 (0.47, 1.18) | 0.215 |
| Undifferentiated Ca | 1.14 (0.72, 1.79) | 0.567 | 1.84 (0.80, 4.29) | 0.152 | 0.67 (0.36, 1.24) | 0.201 |
| Comorbidity | | | | | | |
| No | 0.96 (0.82, 1.11) | 0.556 | 1.75 (1.34, 2.28) | <0.001* | 0.54 (0.44, 0.67) | <0.001* |
| Yes | 1.23 (0.98, 1.54) | 0.080 | 1.43 (0.98, 2.09) | 0.064 | 0.53 (0.39, 0.72) | <0.001* |

SES = Socioeconomic Status, BMI = Body Mass Index.

diseases like cancer is likely to be increased, patients are more likely to accumulates more risk exposure of mortality [18]. Moreover, with advancing age, patients also are more likely to have comorbidities, polypharmacy, physiological changes associated with drug metabolism which can reduce survival [19–21]. Poorer prognosis among lung cancer may reflect the demographic phenomenon, leading to increasing chronic disease mortality with increasing age [22].

Gender did not show any survival differential in our study, which is a stark contrast with existing literature [23,24] many showed men at more risk of mortality compared to women following a diagnosis of lung cancer. One explanation for the indifference may lie in the smaller proportion of females in the study. In the contemporary literature, the impact of gender on survival is far from conclusive though.

In our study, patients with any formal education were found to have a slight survival advantage over those who were illiterate, however, those with the secondary or above level of education didn't show any survival benefit over those who had only a primary level of education. This result may be due to most of the educated patients, are likely to be better-off and preferred treatment elsewhere over public hospitals (where although treatment is free there is significant concern about waiting time, accessibility, and reliability). A very high percentage of illiterate

patients (66.2%) in the public hospital supports the conjuncture. Educational attainment was strongly and inversely associated with mortality. Educational attainment was strongly and inversely associated with mortality from all cancers in population-based observational studies in Unites states [25] and Sweden [26]. However, education does not seem to affect the survival of patients in clinical trials [27]. A possible explanation for such lack of association may be the inherent design structure of clinical trials, where patients in comparing arms are standardized for baseline demography and other characteristics such as education.

With the decrease of socioeconomic status, the odds of both no treatment and nonstandard treatments rise [28,29]. Unavailable or limited health care resources in developing country populations, like Bangladesh, act as a barrier to effective control of incidence and mortality rate. SES continued to exert a small but significant impact on cancer survival, Out-of-pocket healthcare expenditures of households in Bangladesh comprise 64.3% share of the total health expenditure [30]. Due to high out-of-pocket healthcare expenditure and the high cost of treatment, a large number of patients in the lower SES strata are unlikely to complete the treatment or even start the treatment. Economically challenged cancer patients may require special treatment programs that include financial as well as social support.

However, we did not find any survival difference across socioeconomic strata in our study. As around 84% of the patient in the public hospital are from low or lower-middle SES strata, among the rest most middle class. This sample population misses the better-off section of the population, who preferred treatment in private hospitals or in neighboring countries, making a comparison across SES strata implausible.

Low BMI at the time of diagnosis of lung cancer is a consistent marker of poor survival. As a whole, undernutrition is associated with an increased risk of morbidity and mortality associated with non-communicable diseases [31]. Patients with low BMI are more likely to be of advanced stage, hence, the impact on survival among patient with low BMI is probably due to the stage and severity of the disease, rather than BMI itself. Besides, lung cancer patients should undergo an early nutritional evaluation to avoid further deterioration of their nutritional status, which could lead to worse outcomes during treatment as well as survival.

According to Dima et al, lung cancer patients with comorbidity had significantly superior overall survival compared with those without comorbidity. Frequent visits to the physician increase the chance of early diagnosis may be a reason for this outcome [32]. But, with the scarcity of health care resources in a developing country such as Bangladesh, the outcome is different. In this study, we have found the worse survival outcome among those with one or more comorbidity.

One limitation of the study is being a single-centered study however, the study center is the apex public-funded tertiary referral cancer hospital in Bangladesh, where treatments are provided free of cost. As a result, the institution receives patients from all over the country. Patients recruited from the government-funded hospital are likely to exclude a large section of well-off patients of high SES strata, who preferred private hospitals over the public-funded hospital, making the patient population in the study, poorer than the actual Bangladeshi patient population. To keep the selection bias to its minimum and to ensure capturing the spectrum of disease severity, we recruited all the consenting patients with complete data, attended the medical oncology department during the study period. A fair proportion of patients who either refused or were unfit to take treatment or did not finish the treatment were analyzed as no treatment group. Despite being the apex referral cancer center in the country, NICRH is not yet equipped with the facility and capacity newly developed advanced treatment options, only CT and RT were the treatment options available in the hospital, hence this study results may not be relevant to advance and sophisticate, and newly developed treatment options.

In conclusion, older age, presence of one or more comorbidity, poorer performance status, and treatment with RT appeared as a significant predictor of poorer prognosis of lung cancer in Bangladeshi patients. In contrast having, institutional education and treatment with CT and RT combined appeared as a predictor of better prognosis. Treatment with CT and RT combined appear to have better survival for adenocarcinoma and squamous cell carcinoma.

## Acknowledgments

The authors are thankful to Prof. Dr. Qazi Mushtaq Hussain, Director, National Institute of Cancer Research and Hospital, Dhaka, Bangladesh, and his staff, and also to Dr. Ferdous Ara Begum, Dr. Md. Rafiqul Islam, Dr. Asaduzzaman, Dr. Jahangir Alam, Dr. Syeda Masuma Siddiqua, and the faculties & staff of the Department of Medical Oncology, National Institute of Cancer Research and Hospital, Dhaka, Bangladesh for providing the necessary facilities for the preparation of the paper.

## Author Contributions

**Conceptualization:** Muhammad Rafiqul Islam, Md Nazmul Karim.

**Data curation:** Muhammad Rafiqul Islam, A. T. M. Kamrul Hasan, Ishrat Nur Ridi, Md. Mamun Or Rasheed.

**Formal analysis:** Muhammad Rafiqul Islam, Md Nazmul Karim.

**Investigation:** Muhammad Rafiqul Islam.

**Methodology:** Md Nazmul Karim.

**Supervision:** Md Nazmul Karim.

**Validation:** Muhammad Rafiqul Islam.

**Visualization:** Muhammad Rafiqul Islam.

**Writing – original draft:** Muhammad Rafiqul Islam.

**Writing – review & editing:** A. T. M. Kamrul Hasan, Nazrina Khatun, Ishrat Nur Ridi, Md. Mamun Or Rasheed, Syed Mohammad Ariful Islam, Md Nazmul Karim.

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
