## [Decision Letter · Decision Letter 0]

27 Sep 2021

PONE-D-21-18895Demographic differentials of lung cancer survival in Bangladeshi patientsPLOS ONE

Dear Dr. Muhammad Rafiqul Islam,

Thank you for submitting your manuscript to PLOS ONE. After careful consideration, we feel that it has merit but does not fully meet PLOS ONE’s publication criteria as it currently stands. Therefore, we invite you to submit a revised version of the manuscript that addresses the points raised during the review process.

We look forward to receiving your revised manuscript.

Kind regards,

Wen-Wei Sung, M.D., Ph.D.

Academic Editor

PLOS ONE

“No”

Reviewers' comments:

Reviewer's Responses to Questions

**Comments to the Author**

1. Is the manuscript technically sound, and do the data support the conclusions?

Reviewer #1: Partly

Reviewer #2: Yes

Reviewer #3: Yes

Reviewer #4: Partly

2. Has the statistical analysis been performed appropriately and rigorously? 

Reviewer #1: Yes

Reviewer #2: Yes

Reviewer #3: Yes

Reviewer #4: Yes

3. Have the authors made all data underlying the findings in their manuscript fully available?

Reviewer #1: Yes

Reviewer #2: Yes

Reviewer #3: Yes

Reviewer #4: Yes

4. Is the manuscript presented in an intelligible fashion and written in standard English?

Reviewer #1: No

Reviewer #2: Yes

Reviewer #3: Yes

Reviewer #4: No

5. Review Comments to the Author

Reviewer #1: Summary: This is a descriptive manuscript detailing the demographic characteristics and outcomes of patients with a lung cancer diagnosis in the resource-limited country Bangladesh. The authors found that patient with a poor prognosis exhibited older age, presence of one or more comorbidities, a lower performance status and treatment limited to RT only. Prognosis was better in patients with an institutional education and combined treatment with Radiotherapy and Chemotherapy. They conclude that future studies should focus on that subgroup of patients challenged by limited resources to identify novel approaches to improve their prognosis.

MAJOR: This is purely a descriptive study of lung cancer. While classic histology was used to characterize the lung cancer types, it would be worthwhile to know if any mutation analysis was available that could identify genetic drivers of lung cancer. This may well contribute to prognosis irrespective of radiotherapy or chemotherapy. Tobacco exposure seems to be a major factor in lung cancer occurrence. But in cases where it was absent, can the authors speculate on potential non-tobacco related causes? Such as environmental exposure to organic or inorganic dusts, air pollution, smog, indoor smoke exposure from burning fires for cooking. The data are provocative in the sense that it is from a resource limited country where such reporting is often limited so the data are useful to know. It is from a single center although it is a central referral site for cancer evaluation which means people with more advanced disease are likely to be referred so there may be a selection bias toward poorer outcomes. It would be useful to know how decision were made to provide radiotherapy or chemotherapy or both to individual cases. Small cell cancer is treated with chemotherapy and adenocarcinomas are relative resistant to radiotherapy and chemotherapy. Can this be included in the report?

MINOR:

1) English usage and spelling will need to be amended throughout the manuscript. It is unacceptable in its present form.

2) Please clarify the exclusion of patients with targeted therapy immunotherapy. Where there no such patients or was the information on them incomplete so unavailable for study?

3) Squamous cell and small cell cancers are classic tobacco related cancers but adenocarcinomas were the second most common lung cancer type. Was adenocarcinoma more common in woman? Was tobacco exposure more common in men? Any speculation on the frequency of adenocarcinomas?

4) There is little comment on the presence of metastatic disease at the time of diagnosis or a part of therapy. Can this be included?

Reviewer #2: -This is an important study which shows findings in a developing country. It has noted important determinant factor such as education which is important to highlight.

-Education and socio-economic status are closely associated with outcomes even in developed country but in developing country the difference becomes rather stark.

-This study did not find any major difference in mortality based on socioeconomic status was surprising but the reason for that was well explained in the discussion.

Reviewer #3: I greatly appreciate the importance of studying the demographic features of survival in patients with lung cancer, especially since it is a study that involved such a large number of participants.

The article is well written with very few garmmatical errors

Abbreviations must be mentioned at the bottom of the tables

Reviewer #4: This is an interesting observational study looking at how various demographic factors influence lung cancer survival from a public hospital in Bangladesh. The sample size is fairly large for the one year of data accrual (n=1868) and the authors conclude that older age, comorbidities and poorer baseline performance status and use of RT alone as therapy all have poorer prognosis for lung cancer survival. There are a few areas in this study which could benefit from additional analysis, further explanation and some edits to the grammatical style, particularly in the conclusion. This important article will benefit the body of literature regarding lung cancer prognosis, particularly in a poorer/ resource-limited population with more limited access to financial, medical and social resources, a group which may not often be studied.

1) Lack of background explanation on parameter representing performance status.

1a. A major finding of this manuscript is that “poorer performance status” is a significant predictor of poorer prognosis of lung cancer. The authors use the ECOG score to stratify performance status. While they do give the full name for ECOG

[Eastern Cooperative Oncology Group] in the Materials & Methods section, they do not give the full name in the abstract nor in the table 1 legend and they should add it to these areas.

1b. The authors only state that the ECOG score is from 0 to 4 and ultimately placed this cohort int 2 groups with a cut point score of 2 or more. The authors should consider stating what the scores correspond to [i.e., a sore of 0= fully active and score of 4= completely disabled] as not all readers may be familiar with the details of this scoring system.

2) The authors discuss that having a comorbidity also predicts poor outcome for lung cancer but need to give more clear details regarding the comorbidities that they assessed.

2a. In the Materials & Methods section, the authors state, “A patient was considered to have co morbidity if the patient has been suffering or receiving treatment for a major existing condition not related to the cancer or its complication (i.e., diabetes, hypertension, heart disease, or chronic obstructive pulmonary disease).” This sentence is confusing as currently worded. Were the above conditions what was considered a “comorbidity” or were these conditions considered to be related to cancer or its complication ? If these are the diseases being considered then just say that these were considered the comorbidities.

2b. How did the authors determine the co-existence of the above comorbidities: was this based on chart review mentioning one of these disease or was this based on the patient receiving medication to treat one of these diseases ? Please clarify this important point.

2c. It appears that there were 4 comorbidities assessed for (if the above sentence is correct) and the authors appear to have assigned each subject a dichotomous “yes/no” for having a comorbidity. Did the authors do a separate analysis by the number of comorbities for the n=551 who had at least 1 comorbidity. This would be an interesting analysis to see if having 2, 3 or 4 comorbidities conferred an even worse prognosis than just having 1.

3) Tobacco use status could benefit from some additional detail and analysis.

3a. In the Results section, the authors state that, ‘More than eighty percent of patients with lung cancer were smokers and more than half of the patients were smokeless tobacco users.” In table 1, they separate out these 2 different tobacco use categories. There is likely some overlap in the 2 groups which may be a synergistic risk factor (i.e., Both tobacco smoker and smokeless tobacco user). Could the authors provide the information on the number of subjects in the overlap group who used 2 types of tobacco.

3b. Did the authors do a separate analysis on those with both types of tobacco use to see if this was predictive of a poorer outcome ?

3c. Could the authors provide information on the number of patients who were totally non-tobacco exposed (i.e., Not using smoking tobacco or smokeless tobacco) as this information is not able to be determined from Table 1. Was this protective in lung cancer ?

4) Grammatical and typographical errors: Throughout this manuscript there are numerous grammatical and typographical errors which need to be addressed prior to

publication.

4a. Introduction, “…unveil useful insight regarding there link to…” The proper word is “their” not “there.”

4b. Materials & Methods section: There is a missing reference in the following sentence. “Body mass index (BMI) was categorized into …. And > 25 (overweight) (Ref).” Please provide the actual reference here.

4c. Discussion section, in the following sentence “accumulate” should not have an “s.” “…incidence of chronic disease like cancer are likely to be increased, patients are more likely to accumulates more risk exposure.”

4d. Discussion section: I the following sentence “female” should be pleural (i.e., “females”). “…for the indifference may lie in the smaller proportion of female in the study.”

4e. Discussion: in the following sentence the word “who” needs to be added in between “those had.” “…didn’t show any survival benefit over those had only primary level of education...”

4f. Discussion: The following sentence needs to be re-worded. “The impact on survival is probably, correlates more with the stage and severity of the disease, rather than by itself.”

4g. Discussion: There is a typographical error in the sentence below and should read as “218 and 2019.” Please replace the “1” with an “a.” “…attend the hospital during the year 2018 1nd 2019…”

5) Please consider standardizing the abstract.

5a. Statistics presented: For some variables the HR is given while for others no HR is given.

5b. Presentation of results associated with higher survival vs. lower survival. It would be more consistent and send a clearer message to give all of the factors associated with increased survival first and then present the variables associated with lower survival.

6. PLOS authors have the option to publish the peer review history of their article (what does this mean?). If published, this will include your full peer review and any attached files.

Reviewer #1: No

Reviewer #2: **Yes: **Hem Desai

Reviewer #3: **Yes: **khadija AYED

Reviewer #4: No

---

## [Author Response · Author response to Decision Letter 0]

11 Nov 2021

Reviewer 1

MAJOR: This is purely a descriptive study of lung cancer. While classic histology was used to characterize the lung cancer types, it would be worthwhile to know if any mutation analysis was available that could identify genetic drivers of lung cancer. This may well contribute to prognosis irrespective of radiotherapy or chemotherapy. Tobacco exposure seems to be a major factor in lung cancer occurrence. But in cases where it was absent, can the authors speculate on potential non-tobacco related causes? Such as environmental exposure to organic or inorganic dusts, air pollution, smog, indoor smoke exposure from burning fires for cooking. The data are provocative in the sense that it is from a resource limited country where such reporting is often limited so the data are useful to know. It is from a single centre although it is a central referral site for cancer evaluation which means people with more advanced disease are likely to be referred so there may be a selection bias toward poorer outcomes.

 It would be useful to know how decision were made to provide radiotherapy or chemotherapy or both to individual cases. Small cell cancer is treated with chemotherapy and adenocarcinomas are relative resistant to radiotherapy and chemotherapy. Can this be included in the report?

Minor:

1) English usage and spelling will need to be amended throughout the manuscript. It is unacceptable in its present form.

2) Please clarify the exclusion of patients with targeted therapy immunotherapy. Where there no such patients or was the information on them incomplete so unavailable for study?

3) Squamous cell and small cell cancers are classic tobacco related cancers but adenocarcinomas were the second most common lung cancer type. 

Was adenocarcinoma more common in woman? Was tobacco exposure more common in men? Any speculation on the frequency of adenocarcinomas?

4) There is little comment on the presence of metastatic disease at the time of diagnosis or a part of therapy. Can this be included?

Response:-1(Major)

Many thanks for the comment. We agree with the reviewer about the importance of the mutation analysis. Unfortunately, National Institute of Cancer Research and Hospital (NICRH), despite being the apex referral cancer centre in the country, is not yet equipped with the facility and capacity of mutation analysis, hence we could not provide any insight in this regard. 

We have included the following text in the discussion section of the manuscript accordingly

Being single centre is a limitation of the study, we acknowledged this as limitation. Regarding selection bias the reviewer have raised a very legitimate and important concern. However, the main factor that would contribute to the selection bias is the socioeconomic status, rather than the stage of the disease. 

“Patients recruited from the government-funded hospital are likely to exclude a large section of well-off patients of high SES strata, who preferred private hospitals over the public-funded hospital, making the patient population in the study, poorer than the actual Bangladeshi patient population. To keep the selection bias to its minimum and to ensure capturing the spectrum of disease severity, we recruited all the consenting patients with complete data, attended the medical oncology department during the study period.” 

 Regarding choice of treatment, we agree with reviewers’ comment. Following text has been added in the method section regarding choice of treatment.

“After complete assessment of the patient, Histology based standard chemotherapy and radiotherapy protocols guided by the National Comprehensive Cancer Network (NCCN) guidelines were followed by an institutional multidisciplinary tumor board (comprise medical Oncologist, Radiation oncologist, surgical oncologist, radiologist and pathologist) for both curative and palliative setting. 

Response -1(Minor)

Thank you so much for the comment.

1.We have now made our best effort to review the English usage and spelling.

2. We acknowledge, our patient population did not include patient with targeted therapy. Unfortunately, National Institute of Cancer Research and Hospital (NICRH), despite being the apex referral cancer centre in the country, is not yet equipped with the facility and capacity of several targeted therapy, immunotherapy in particular. We have added the following text in the revised manuscript to address reviewer’s comment. 

“despite being the apex referral cancer centre in the country, NICRH is not yet equipped with the facility and capacity of targeted therapy, immunotherapy in particular. As a result, our patient population did not include patients with targeted therapy (i.e. immunotherapy).

3. Thanks for the insight, we have revealed the following information from the data and added relevant text in the discussion section of the revised manuscript. 

Prevalence of Adenocarcinoma is higher in female (46.3%) than male (36.9%), and conversely the prevalence of Squamous cell carcinoma is higher in male (43.2%) than female (37.5). The finding supports the classical conjuncture of squamous cell lung cancer being related to tobacco exposure. Gender specific tobacco exposure also support the fact, as tobacco consumption was higher among males (85.4%) in comparison to female (55.6%).

4.Thanks for the comment, this is important, and we have included the following text in the result section. 

“About 19.4% patients had metastatic disease during the presentation. A third of those patients had liver metastasis.” 

Reviewer #2 

This is an important study which shows findings in a developing country. It has noted important determinant factor such as education which is important to highlight.

-Education and socio-economic status are closely associated with outcomes even in developed country but in developing country the difference becomes rather stark.

-This study did not find any major difference in mortality based on socioeconomic status was surprising but the reason for that was well explained in the discussion.

Response-2 

Many thanks for your comment. This is a huge encouragement for us. 

Reviewer #3 

I greatly appreciate the importance of studying the demographic features of survival in patients with lung cancer, especially since it is a study that involved such a large number of participants.

The article is well written with very few grammatical errors

Abbreviations must be mentioned at the bottom of the tables 

Response-3

Thanks for the valuable comments.

1. We put our best effort to correct the grammatical errors 

2. We have added the abbreviations at the bottom of the tables in the revised manuscript. 

Reviewer #4

1) Lack of background explanation on parameter representing performance status.

1a. The authors use the ECOG score to stratify performance status. While they do give the full name for ECOG

[Eastern Cooperative Oncology Group] in the Materials & Methods section, they do not give the full name in the abstract nor in the table 1 legend and they should add it to these areas.

1b. The authors only state that the ECOG score is from 0 to 4 and ultimately placed this cohort int 2 groups with a cut point score of 2 or more. The authors should consider stating what the scores correspond to [i.e., a sore of 0= fully active and score of 4= completely disabled] as not all readers may be familiar with the details of this scoring system.

2) The authors discuss that having a comorbidity also predicts poor outcome for lung cancer but need to give more clear details regarding the comorbidities that they assessed.

2a. In the Materials & Methods section, the authors state, “A patient was considered to have co morbidity if the patient has been suffering or receiving treatment for a major existing condition not related to the cancer or its complication (i.e., diabetes, hypertension, heart disease, or chronic obstructive pulmonary disease).” This sentence is confusing as currently worded. Were the above conditions what was considered a “comorbidity” or were these conditions considered to be related to cancer or its complication? If these are the diseases being considered then just say that these were considered the comorbidities.

2b. How did the authors determine the co-existence of the above comorbidities: was this based on chart review mentioning one of these diseases or was this based on the patient receiving medication to treat one of these diseases? Please clarify this important point.

2c. It appears that there were 4 comorbidities assessed for (if the above sentence is correct) and the authors appear to have assigned each subject a dichotomous “yes/no” for having a comorbidity. Did the authors do a separate analysis by the number of comorbidities for the n=551 who had at least 1 comorbidity. This would be an interesting analysis to see if having 2, 3 or 4 comorbidities conferred an even worse prognosis than just having 1.

3) Tobacco use status could benefit from some additional detail and analysis.

3a. In the Results section, the authors state that, ‘More than eighty percent of patients with lung cancer were smokers and more than half of the patients were smokeless tobacco users.”

 In table 1, they separate out these 2 different tobaccos use categories. There is likely some overlap in the 2 groups which may be a synergistic risk factor (i.e., Both tobacco smoker and smokeless tobacco user). Could the authors provide the information on the number of subjects in the overlap group who used 2 types of tobacco.

3b. Did the authors do a separate analysis on those with both types of tobacco use to see if this was predictive of a poorer outcome?

3c. Could the authors provide information on the number of patients who were totally non-tobacco exposed (i.e., Not using smoking tobacco or smokeless tobacco) as this information is not able to be determined from Table 1. Was this protective in lung cancer?

4) Grammatical and typographical errors: Throughout this manuscript there are numerous grammatical and typographical errors which need to be addressed prior to

publication.

4a. Introduction, “…unveil useful insight regarding there link to…” The proper word is “their” not “there.”

4b. Materials & Methods section: There is a missing reference in the following sentence. “Body mass index (BMI) was categorized into …. And > 25 (overweight) (Ref).” Please provide the actual reference here.

4c. Discussion section, in the following sentence “accumulate” should not have an “s.” “…incidence of chronic disease like cancer are likely to be increased, patients are more likely to accumulates more risk exposure.”

4d. Discussion section: I the following sentence “female” should be pleural (i.e., “females”). “…for the indifference may lie in the smaller proportion of female in the study.”

4e. Discussion: in the following sentence the word “who” needs to be added in between “those had.” “…didn’t show any survival benefit over those had only primary level of education...”

4f. Discussion: The following sentence needs to be re-worded. “The impact on survival is probably, correlates more with the stage and severity of the disease, rather than by itself.”

4g. Discussion: There is a typographical error in the sentence below and should read as “218 and 2019.” Please replace the “1” with an “a.” “…attend the hospital during the year 2018 1nd 2019…”

5) Please consider standardizing the abstract.

5a. Statistics presented: For some variables the HR is given while for others no HR is given.

5b. Presentation of results associated with higher survival vs. lower survival. It would be more consistent and send a clearer message to give all of the factors associated with increased survival first and then present the variables associated with lower survival.

Response-4

Thanks for your valuable comment. 

2a. The wording of the sentence we presented left the ground for ambiguity. We have rewritten the section to be clearer. 

Following text has been added to the methods section of the revised manuscript 

“A patient was considered to have co-morbidity if the patient had been suffering from or receiving treatment for a major existing condition such as, diabetes, hypertension, heart disease, or chronic obstructive pulmonary disease etc.”

2b. We agree with the author that, this is an important point. The following text has been added the method section of the revised manuscript. 

“Comorbidity data of the patient were collected by trained data collector (physician) from the, chart review, current medication record, and cancer treatment eligibility check-up record conducted at the center.”

2c. Thanks for the insightful comment, we have conducted additional analysis according to the comment. We have added the following result in the result section of the revised manuscript. 

“Around 29.5% (n=551) patients were reported to have at least one comorbidity, the comorbidities were Diabetes (9.0%), Hypertension (10.1%), COPD (11.9%), Asthma (1.0%), IHD (0.9%), Arthritis (1.7%). Out of the 551 patients with comorbidities, 461 had one comorbidity, 82 had two comorbidities and 7 patients had three or more comorbidities. Although presences of at least one co-morbidity is associated with lower survival. However, having multiple co-morbidity did not exert any additional synergistic effect on survival.”

Thanks for the insightful comment and we agree with the reviewer. We ran additional analysis and the following texts are added in the result section of the revised manuscript. 

3a. “44.3% (n=827) patients consumed both the form of tobacco”. 

3b. Consumption of any form of tobacco neither individual nor both are predictive of survival. 

3c. “10.8% (208) persons were neither smoke tobacco nor smokeless tobacco user, their tobacco consumption status also did not seem to affect the survival” 

4. Thank you so much for your thoughtful comments. We have corrected the following as suggested. 

4a. “their”

4b. reference added. 

4c. accumulate 

4d. Females “s” added

4e. “who” added 

4f. “Patients with low BMI are more likely to be of advanced stage, hence, the impact on survival among patient with low BMI is probably due to the stage and severity of the disease, rather than BMI itself.”

4g. and

5 (a.b). Thank you so much for your thoughtful comments. We have rewritten the abstract to standardize it as suggested, further the following texts have been included in the abstract of the revised manuscript

“Higher survival was associated with institutional education (HR 0.9; 95% CI 0.77, 0.99), and receipt of combined radiotherapy and chemotherapy (HR 0.56; 95% CI 0.46, 0.65; p <0.001). In contrast, lower survival was associated with older age between 60- 69 years (HR 1.3; 95% CI 1.3, 1.5;), age ≥ 70 years (HR 1.4; 95% CI 1.1, 1.7), having any comorbidity (HR 1.1; 95% CI 1.0, 1.3), with ECOG score ≥ 3 (HR 1.41; 95% CI 1.01, 1.96) and receipt of radiotherapy treatments only (HR 1.6; 95% CI 1.3, 1.9).”

---

## [Decision Letter · Decision Letter 1]

26 Nov 2021

Demographic differentials of lung cancer survival in Bangladeshi patients

PONE-D-21-18895R1

Dear Dr. Muhammad Rafiqul Islam,

We’re pleased to inform you that your manuscript has been judged scientifically suitable for publication and will be formally accepted for publication once it meets all outstanding technical requirements.

Kind regards,

Wen-Wei Sung, M.D., Ph.D.

Academic Editor

PLOS ONE

Reviewers' comments:

Reviewer's Responses to Questions

**Comments to the Author**

1. If the authors have adequately addressed your comments raised in a previous round of review and you feel that this manuscript is now acceptable for publication, you may indicate that here to bypass the “Comments to the Author” section, enter your conflict of interest statement in the “Confidential to Editor” section, and submit your "Accept" recommendation.

Reviewer #1: All comments have been addressed

2. Is the manuscript technically sound, and do the data support the conclusions?

Reviewer #1: Yes

3. Has the statistical analysis been performed appropriately and rigorously? 

Reviewer #1: Yes

4. Have the authors made all data underlying the findings in their manuscript fully available?

Reviewer #1: Yes

5. Is the manuscript presented in an intelligible fashion and written in standard English?

Reviewer #1: Yes

6. Review Comments to the Author

Reviewer #1: The authors have now addressed the concerns raised in the initial review. The revision is acceptable for publication.

7. PLOS authors have the option to publish the peer review history of their article (what does this mean?). If published, this will include your full peer review and any attached files.

Reviewer #1: No

---

## [Editor Report · Acceptance letter]

3 Dec 2021

PONE-D-21-18895R1 

Demographic differentials of lung cancer survival in Bangladeshi patients 

Dear Dr. Islam:

I'm pleased to inform you that your manuscript has been deemed suitable for publication in PLOS ONE. Congratulations! Your manuscript is now with our production department. 

Kind regards, 

on behalf of

Dr. Wen-Wei Sung 

Academic Editor

PLOS ONE